

# Detection of wakes in the inflow of turbines using nacelle lidars

Dominique P Held[1,2] and Jakob Mann[1]

[1]Department of Wind Energy, Technical University of Denmark (DTU), Frederiksborgvej 399, 4000 Roskilde, Denmark
[2]Windar Photonics A/S, Helgeshøj Alle 16-18, 2630 Taastrup, Denmark

*Correspondence to:* Dominique P Held (domhel@dtu.dk)

**Abstract.** Nacelle-mounted lidar systems offer the possibility of remotely sensing the inflow of wind turbines. Due to the limitation of line-of-sight measurements and the limited number of focus positions, assumptions are necessary to derive useful inflow characteristics. Typically, horizontally homogeneous inflow is assumed which is well satisfied in flat, homogeneous terrain and over sufficiently large time averages. However, it is violated if a wake impinges the field of view of one of the beams. In such situations, the turbine yaw misalignment measurements show large biases which require the detection and correction of these observations. Here, a detection algorithm is proposed based on the spectral broadening of the Doppler spectrum due to turbulence within the probe volume. The small-scale turbulence generated within wake flows will typically lead to a significantly larger broadening than in the ambient flow. Thus, by comparing the spectral widths at several locations, situations, where a wake is impinging the field of view of one or more beams can be identified. The correction method is based on an empirical relationship between the difference in turbulence levels at distinct beams and the difference in wind direction derived from the lidar and the real wind direction. The performance of the algorithm is evaluated in a field experiment identifying all wake situations, and thus, correcting the lidar derived wind direction.

## 1 Introduction

Modern wind turbines usually follow an upwind turbine design where the rotor is installed upstream of the tower. This design requires an active yaw angle control which traditionally uses nacelle-mounted sonic anemometer or wind vane misalignment measurements to align the turbine with the wind. The yaw motor actuation is slow to avoid excessive wear on its components and temporary yaw misalignment is unavoidable (Burton et al., 2011). On the other hand, a systematic error in yaw tracking leads to production losses and should be prevented.

One source of yaw tracking error stems from the misalignment measurements provided by the nacelle sonic anemometer or wind vane which are heavily disturbed by the flow around the rotor blades and the nacelle. These yaw sensors can be calibrated against undisturbed reference wind direction sensors, for example mast mounted sonic anemometers and wind vanes or vertically profiling lidars. However, differences in inflow and site conditions between calibration site and actual installation site of the turbine are not reflected in this calibration and might introduce biases in the measurements. A method to detect and correct the degradation of the sensors over time is presented in Mittelmeier and Kühn (2018). Additionally, field experiments showed that wakes from neighboring turbines can lead to yaw offsets from the dominant wind direction for downstream turbines (Schepers, 2009; McKay et al., 2013).





Several alternative sensors have been proposed to sense yaw misalignment. In Pedersen et al. (2015) the use of spinner anemometry is suggested to overcome the flow distortions from blades and the nacelle. Three 1-dimensional sonic anemometers were mounted at the spinner to measure horizontal wind speed, yaw misalignment and flow inclination. Calibration of the sensor is necessary to remove measurement biases (Demurtas and Janssen, 2016).

A wind state observer for vertical and horizontal shear, yaw misalignment and flow inclination based on flap- and edgewise blade bending moment measurements was formulated and tested through simulations in Bertelè et al. (2017) and validated on a scaled experimental turbine in Bertelè et al. (2018). In Bottasso et al. (2018) a similar observer using flapwise blade bending moments was developed to estimate the average wind speed over four azimuthal sector of the rotor plane. By comparing upper with lower and left with right sector vertical shear and impinging wakes from upstream turbines could be detected. For the

detection of wakes the derived sector-wise mean wind speed and turbulence intensity were used.

Nacelle-mounted light detection and ranging (lidar) devices have also been suggested to estimate yaw misalignment (Scholbrock et al., 2015; Fleming et al., 2014). Due to their remote sensing capabilities, it is possible to measure the inflow in front of the turbine. Measurements at close distance from the rotor are affected by the induction of the turbine but the severe influences of blades and nacelle behind the rotor can be avoided. However, the limitation of measuring only the line-of-sight (LOS) com-

ponent along the laser beam requires the lidar systems to probe the wind field at several position over the rotor plane and to make assumptions of the incoming flow field to derive the yaw misalignment of the turbine. As a consequence, nacelle lidars are able to steer the laser light towards several focus positions. The most simple setup is a 2-beam system which scans two horizontal positions at the hub height of the turbine, see fig. 1.

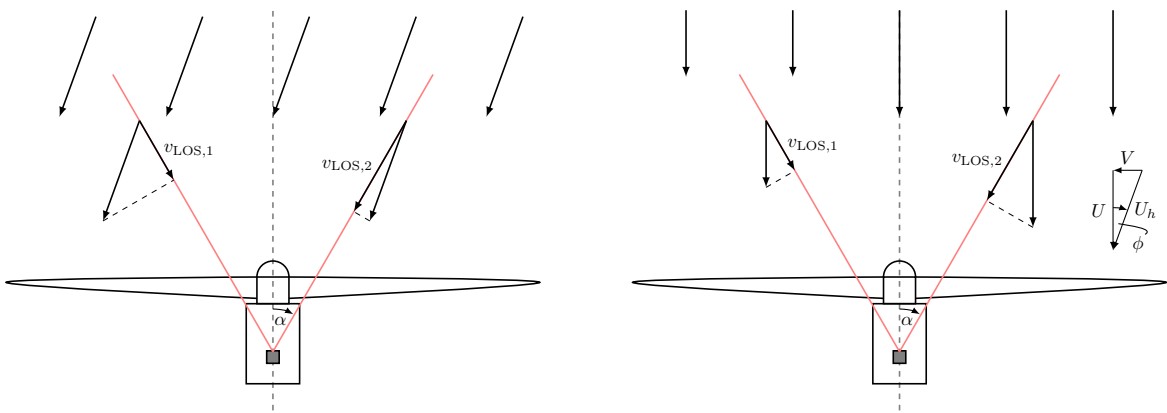

**Figure 1.** Illustration of a 2-beam nacelle-lidar mounted on a wind turbine to measure yaw misalignment. Misaligned horizontally homogeneous flow will affect the LOS measurements at the two focus positions (*left*). A wake in the field of view on one of the beams will affect the LOS measurement in a similar way as a misaligned flow (*right*). From lidar measurements these two situations can not be distinguished.

To derive the yaw misalignment from nacelle lidar measurements two common assumptions are made:

1.  No vertical wind vector component



### 2. Horizontally homogeneous flow

In case of a horizontally scanning lidar the measured vertical component will always be very small since only the LOS can be measured. Further, the horizontally homogeneous flow assumption is well satisfied if the terrain is flat and homogeneous and if appropriate averages over time (commonly one to 10 minute averages) are taken. Note that by changing the second assumption

to a perfect yaw alignment, the measurement of the lidar system can be interpreted as horizontal shear. This can pose a problem if the lidar is used for yaw alignment and cyclic pitch control (Schlipf, 2015).

The assumption of horizontal homogeneity is violated if a wake from an upstream turbine is impinging the field of view of one of the beams. Such a situation is illustrated in the right panel of fig. 1. The left half of the rotor is exposed to a lower wind speed due to the wake and thus the measurement at the left LOS, $v_{\mathrm{LOS},1}$, is reduced. This is similar to a reduction of one of the

LOS speed due to misaligned flow and cannot be distinguish from lidar measurements, see left illustration in fig. 1. Thus, an impinging wake in the field of view of one of the beams will lead to a yaw misalignment measurement even though the flow is aligned. If the lidar should supply yaw misalignment measurements to the yaw controller, these situations need to be identified and, if possible, the influence of the wake needs to be corrected for.

In general, scanning lidar systems, where the laser follows a trajectory rather than probes discrete points, are able to measure

large scale flow phenomena due to their high spatial and temporal resolution. The first field experiment of measuring wakes behind turbines by a scanning continuous-wave (cw) lidar was conducted by Bingöl et al. (2010) and Trujillo et al. (2011). A cw lidar was mounted on the nacelle of a turbine looking downstream and the derived flow information was sufficient to identify and and characterize the meandering of the wake. More recently, a nacelle-mounted cw lidar with a fast scanning head was used to provide a detailed image of wake behind a turbine and track its movement under different atmospheric conditions

(Herges et al., 2017). Long-range pulsed lidars positioned on the ground (Iungo et al., 2013; Smalikho et al., 2013; Bodini et al., 2017) or in offshore wind farms (Krishnamurthy et al., 2017) have also been used to measure wakes. The advantage of pulsed systems is their increased range (of several kilometers) compared to cw systems which allows to visualize the wakes of multiple turbines in a wind farm.

However, the limited amount of information gathered by commercially available nacelle lidars makes the detection and

characterization of wakes challenging. Currently, commercially available systems include Windar Photonics' 2- and 4-beam cw lidars with fixed focus distance, Leosphere's 4-beam pulsed lidar and ZX Lidars' cw lidar with 50 focus points distributed on a cone at variable focus distances[1].

In this study Windar Photonics' 2-beam nacelle lidar will be used to investigate the effect of wakes on the turbine yaw misalignment measurements. The aim is to detect half-wake situations that lead to biased misalignment measurements and

correct the measurements so that they can be used to perform turbine yaw alignment. First, we will propose an algorithm that utilizes the spectral broadening of Doppler peaks from small-scale turbulence to detect wakes in the turbine inflow (sec. 2). We will then present the results of the algorithm (sec. 4) gathered during a field experiment described in sec. 3. Finally, an empirical relationship between the spectral width measurements and the difference in wind direction between lidar and a mast-mounted sonic anemometer is used to correct the erroneous misalignment measurements.

---

[1]More information can be found on the company websites: Windar Photonics A/S, Leosphere and ZX Lidars.





## 2 Methodology

### 2.1 Measurement principles of nacelle lidars

The fluctuating part of a 3-dimensional (3D) wind field can be described by the vector field $\boldsymbol{u}(\boldsymbol{x},t) = (u_1, u_2, u_3)$, where $\boldsymbol{x}$ and $t$ refer to a position in space and time, respectively. Lidar systems can sense wind speed by measuring the Doppler frequency

shift of the backscattered laser light from aerosols suspended in the atmosphere. Cw systems emit light focused on a point in space continuously, while pulsed systems transmit parcels of light and measure the backscatter as the light progresses in space. In this study we will focus on cw lidars.

The measurement process of a cw lidar can be represented mathematically as the convolution between the projected LOS component $\boldsymbol{n} \cdot \boldsymbol{u}$ and a weighting function $\varphi(s)$ based on the laser intensity:

$$v_{\text{LOS}}(\boldsymbol{r}) = \int\limits_{-\infty}^{+\infty} \varphi(s)\boldsymbol{n} \cdot \boldsymbol{u}(s\boldsymbol{n} + \boldsymbol{r})ds, \tag{1}$$

where $\boldsymbol{n}$ is the laser beam unit directional vector, $\boldsymbol{r}$ is the focus position and $s$ is distance from the focus position along the laser line (Mann et al., 2010). The weighting function of a cw lidar can be approximated by (Sonnenschein and Horrigan, 1971)

$$\varphi(s) = \frac{1}{\pi} \frac{z_R}{z_R^2 + s^2}, \tag{2}$$

where $z_R$ is the Rayleigh length that characterizes the probe volume of the lidar. It depends on the focus distance squared,

i.e. $z_R \propto |\boldsymbol{r}|^2$, and can vary between a few centimeters to tens of meters. The probe volume has an attenuating effect on the turbulent fluctuations in $\boldsymbol{u}$ and variances (and thus turbulence intensities) will be underestimated. The attenuated turbulent fluctuations will cause a broadening of the Doppler spectrum. The Doppler spectrum $S(v, \boldsymbol{r})$ as function of LOS component wind speed can be defined by

$$S(v, \boldsymbol{r}) = \int\limits_{-\infty}^{\infty} \varphi(s)\delta(v - \boldsymbol{n} \cdot \boldsymbol{u}(s\boldsymbol{n} + \boldsymbol{r}))ds, \tag{3}$$

where $\delta(.)$ is the Dirac delta function. The delta function implies that the Doppler spectrum is an integration (or summation) of the values of $\varphi(s)$, where $\boldsymbol{n} \cdot \boldsymbol{u}(s\boldsymbol{n} + \boldsymbol{r}) = v$. The LOS speed $v_{\text{LOS}}$, as defined by eq. 1, is the first statistical moment of $S(v, \boldsymbol{r})$:

$$v_{\text{LOS}}(\boldsymbol{r}) = \int\limits_{-\infty}^{\infty} vS(v, \boldsymbol{r})dv, \tag{4}$$

where $\int_{-\infty}^{\infty} S(v, \boldsymbol{r})dv = 1$ is assumed. The second central statistical moment of $S(v, \boldsymbol{r})$ is the variance and a measure of the

width of $S(v, \boldsymbol{r})$:

$$\sigma_{\text{LOS}}^2(\boldsymbol{r}) = \int\limits_{-\infty}^{\infty} (v - v_{\text{LOS}}(\boldsymbol{r}))^2 S(v, \boldsymbol{r})dv. \tag{5}$$



Based on the inflow assumption mentioned in sec. 1, the LOS speed measurements at the two positions of a 2-beam lidar can be used to derive the yaw misalignment of the turbine (see also fig. 1). The following set of equations are used, where the hat notation ($\widehat{\cdot}$) is used to indicate estimations from the lidar:

$$v_{\mathrm{LOS},1} = U\cos\alpha - V\sin\alpha \tag{6}$$

$$v_{\mathrm{LOS},2} = U\cos\alpha + V\sin\alpha \tag{7}$$

$$\widehat{U} = \frac{v_{\mathrm{LOS},2} + v_{\mathrm{LOS},1}}{2\cos\alpha} \tag{8}$$

$$\widehat{V} = \frac{v_{\mathrm{LOS},2} - v_{\mathrm{LOS},1}}{2\sin\alpha} \tag{9}$$

$$\widehat{U}_{\mathrm{h}} = \sqrt{\widehat{U}^2 + \widehat{V}^2} \tag{10}$$

$$\widehat{\phi} = \tan^{-1}\left(\frac{\widehat{V}}{\widehat{U}}\right), \tag{11}$$

where $U$ and $V$ are longitudinal and lateral wind speed, respectively, and $\alpha$ is the angle between the shaft axis and the laser beam. In the case of the 2-beam lidar by Windar Photonics, $\alpha = 30°$. Yaw misalignment is defined as the angle the turbine needs to yaw in the clockwise direction (seen from above) to align itself with the flow (see fig. 1).

An impinging wake in the field of view of one of the beams will lead to a bias in the lidar estimated quantities (eqs. 8 to 11). Here, we will analyze the case, where LOS 1 is affected by a reduction in the streamwise wind speed component due to an upstream wake deficit $u_{\mathrm{w}}$, i.e. $v_{\mathrm{LOS},1} = (U - u_{\mathrm{w}})\cos\alpha - V\sin\alpha$. The influence of $u_{\mathrm{w}}$ on $\widehat{\phi}$ and $\widehat{U}_{\mathrm{h}}$ for different turbine misalignments $\phi$ is presented in fig. 2.

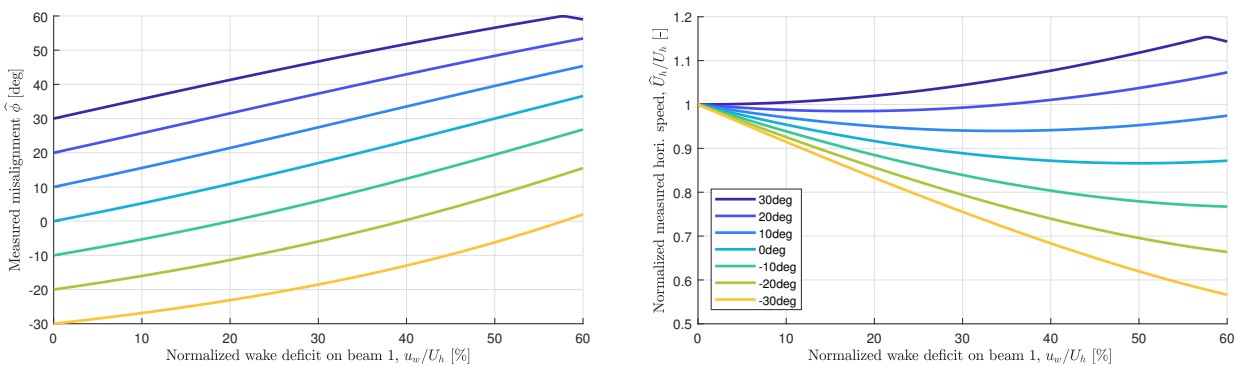

**Figure 2.** Influence of a streamwise flow speed reduction due to a wake $u_{\mathrm{w}}$ at LOS 1 on the lidar estimates of turbine yaw misalignment $\widehat{\phi}$ and horizontal wind speed $\widehat{U}_{\mathrm{h}}$. The differently colored lines represent different turbine misalignments $\phi$.

It can be seen that an increasing wake deficit imposes a positive bias onto $\widehat{\phi}$ since $v_{\mathrm{LOS},1}$ is reduced. The effect is strongest for negative turbine misalignments, where the flow is aligned more towards LOS 1. The kinks that appear for the negative turbine misalignments occur once the wake deficit reduces the LOS 1 component to negative values. However, since the lidar uses a homodyne detection method only the magnitude, not the sign, of LOS 1 component can be sensed and the measured



misalignment ranges between $-60°$ and $60°$. The effect of $u_{\mathrm{w}}$ on $\widehat{U}_{\mathrm{h}}$ leads to an over- and underestimation of the horizontal wind speed. Again, the negative turbine misalignments are effected more severely than positive misalignments.

## 2.2 Wake Detection Algorithm

The wake detection algorithm is based on the following two principles:

1. Within wind turbine wakes turbulence with a smaller length scale than the ambient flow is generated.

2. The small-scale turbulence will lead to a broadening of the Doppler spectrum due to the large probe volume of the lidar system.

The generation of small-scale turbulence has been investigated through simulations, scaled wind tunnel experiments and field tests. The primary aim was to validate mean wake deficit profiles, but some studies also investigated turbulence profiles and
applied spectral analysis.

For example, in Troldborg et al. (2007) large eddy flow simulations were conducted with an actuator line model representing the wind turbine. Enhanced levels of small-scale turbulence inside the wake were found. It was also found that the turbulence inside a wake is more isotropic than the ambient flow. Similar results were found in scaled wind tunnel test. Singh et al. (2014) showed that wind turbines extract energy from mean and large-scale structures while increasing small-scale turbulence. In the
near-wake a clear influence of the rotor is visible (e.g spikes at tower and blade frequencies in the turbulence spectra), but in the far-wake these effects merged into a range of amplified small-scale turbulence (Chamorro et al., 2012).

Field experiments using mast anemometry also compared power spectra between wake and ambient flow. In Højstrup (1999) input of energy to high frequencies was observed, which was detectable up to 14.5 rotor diameters behind the turbine. Similar result were obtained in Högström et al. (1988) where the high frequency components of the streamwise component increased
fourfold in the wake. Iungo et al. (2013) analyzed wake observations gathered with a pulsed lidar system and also found increased turbulence, but more turbulence was created at the top of the rotor compared to the bottom part.

The effect spectral broadening of the Doppler spectrum due to turbulence and Doppler spectrum width measurements has been considered in several studies, for an overview see Sathe and Mann (2013). For example, Smalikho (1995) described the broadening process theoretically and proposed a method to measure the dissipation rate of turbulent kinetic energy. Branlard
et al. (2013) showed that long time averages of the Doppler spectrum (in this case 10 and 30 minutes) approaches the probability density function of a sonic anemometer and can be used to improve variance measurements. The same method to derive unfiltered variance was used for vertical profiling lidars (Mann et al., 2010) and nacelle lidars (Peña et al., 2017).

From both properties mentioned above, we will define the LOS-equivalent turbulence intensity, $\mathrm{TI}_{\mathrm{LOS}}$, to characterize the small-scale turbulence contained in the probe volume of the lidar:

$$\mathrm{TI}_{\mathrm{LOS}}(\boldsymbol{r}) = \frac{\sigma^2_{\mathrm{LOS}}(\boldsymbol{r})}{v_{\mathrm{LOS}}(\boldsymbol{r})} \tag{12}$$

and the difference in $\mathrm{TI}_{\mathrm{LOS}}$ between LOS 1 and LOS 2:

$$\Delta\mathrm{TI}_{\mathrm{LOS}} = \mathrm{TI}_{\mathrm{LOS}}(\boldsymbol{r}_{\mathrm{LOS1}}) - \mathrm{TI}_{\mathrm{LOS}}(\boldsymbol{r}_{\mathrm{LOS2}}), \tag{13}$$





where the $r_{\text{LOS1}}$ and $r_{\text{LOS2}}$ refer to the focus positions of beam 1 and 2. The turbulence intensities and their differences will be calculated from one minute average spectra. If a beam is exposed to a wake inflow we expect an increased $\text{TI}_{\text{LOS}}$ compared to a beam exposed to the ambient flow. Considering turbulence intensities allows to compare different wind speed regimes.

The wake detection algorithm is designed to treat the one minute spectra consecutively and will compare the values of the detection parameters $\text{TI}_{\text{LOS}}(r_{\text{LOS},1})$, $\text{TI}_{\text{LOS}}(r_{\text{LOS},2})$, $\Delta\text{TI}_{\text{LOS}}$ and $\widehat{\phi}$ to their values from ambient, wake-free conditions. Based on threshold values on the detection parameters the algorithm will then decide whether one or both beams are affected by a wake. For example, if beam 1 is affected by a wake, $v_{\text{LOS},1}$ will be reduced compared to $v_{\text{LOS},2}$ even though the misalignment of the turbine will not change. Simultaneously, the turbulence intensity on beam 1 will increase due to the small-scale turbulence within the probe volume and a detection signal will be visible on $\Delta\text{TI}_{\text{LOS}}$. The influence of the wake can also be seen on the

yaw misalignment estimation $\widehat{\phi}$ which will deviate from its mean value in ambient flow. A reversed situation appears when a wake is affecting beam 2. At the initialization the algorithm requires some observations to establish correct values of the running averages. From our experience a few hours of wake-free data is sufficient.

## 3   Experimental setup

In this study, two field experiments executed at DTU's test site at the Risø Campus were analyzed. The site consists of a row

of turbines, of which two have been operative during the experiment: a Vestas V52 turbine with 850 kW rated power, a hub height of 44 m and a diameter of 50 m and a smaller Nordtank turbine with 500 kW rated power, a hub height of 36 m and a diameter of 41 m. Further, data from a meteorological mast at a distance of $120\,m = 2.3 D_{\text{V}}$ from the Vestas V52 turbine, where $D_{\text{V}} = 52$ m. The distance between the Vestas and the Nordtank turbine is 215 m ($5.2 D_{\text{N}}$ or $4.1 D_{\text{V}}$) at an angle of $195°$ (clockwise from north), where $D_{\text{N}} = 41$ m. A picture of the site can be found in fig. 3.

The Vestas V52 turbine was equipped with a 2-beam nacelle lidar by Windar Photonics during two distinct periods. Initially, one system was mounted between 5th December 2015 and 12th January 2016 and a second system was mounted between 29th March 2016 and 4th May 2016. The systems have identical properties. The opening angle between shaft axis and beam was $30°$ and the focus distance was 37 m (implying that $z_R = 2.1$ m). The objective of the experiment was to obtain yaw misalignment measurements from the lidars with special focus on the influence of the wake from the Nordtank turbine onto the Vestas V52.

The data acquisition system of the turbine was logging turbine, mast and lidar data at a sampling rate of 35 Hz. For the analysis the data has been downsampled to 1 Hz.

The wind rose during the two experimental periods derived from a sonic anemometer mounted at the Vestas V52 hub height on the meteorological mast is shown in fig. 3. The direction of wakes from Nordtank onto V52 is indicated as a dashed black line. Two dominant direction can be identified; the waked wind direction only represents a small share of the data.





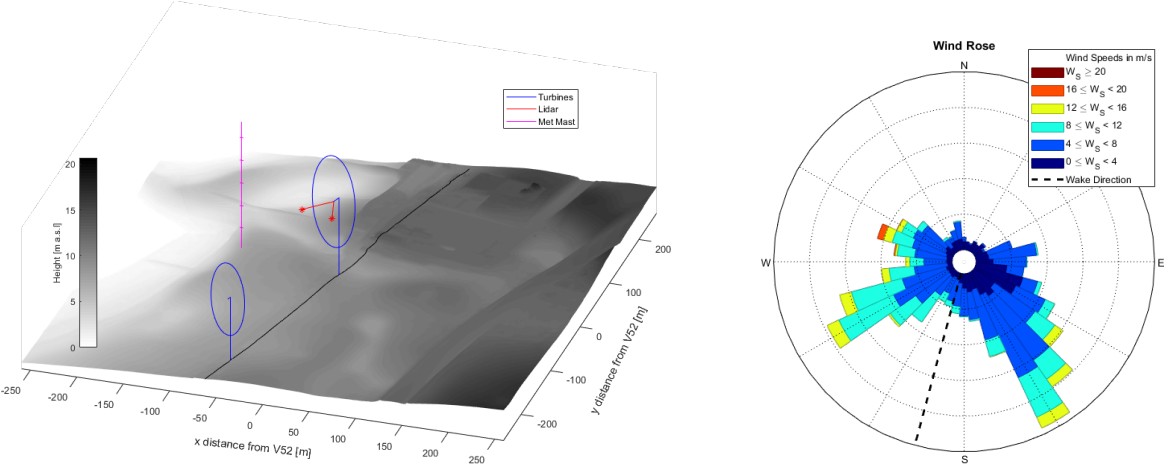

**Figure 3.** *Left*: Digital terrain model (DTM) of DTU's test site at Risø where the two turbines and the lidar are also shown. The black line indicates the wake direction of 195°. Zone 32 UTM coordinates centered at the Vestas V52 turbine were used. The DTM data was obtained from the Danish Map Supply (Agency for Data Supply and Efficiency). *Right*: Wind rose derive from sonic anemometer measurements on the meteorological mast during the experiment. The dashed black line indicates the direction of the Nordtank turbine.

## 4 Results

In this study data is compared during normal power production of the turbines. To achieve this, lower thresholds on the minimum power production (i.e. $> 0\,\mathrm{kW}$), minimum rotor speed (i.e. $> 16\,\mathrm{rpm}$) and maximum pitch angle (i.e. $< 23°$) in each 10 minute period were applied for the Vestas V52 turbine. Since the Nordtank turbine is stall regulated a lower threshold on minimum power production (i.e. $> 0\,\mathrm{kW}$) and minimum rotor speed (i.e. $> 27\,\mathrm{rpm}$) have been applied. The filtering removed
53.3% of the data for both experiments. Additionally a filter was applied on the lidar data. Instances where the LOS speed could not be determined from the Doppler spectra have been summed within a 10 minute period. If more than 10% of the data (or 60 measurements) was missing, the 10 minute period was rejected. Instances where four or more consecutive unavailable measurements occurred on any of the two beams were also discarded. Whether a measurement is available or not was decided
internally by the lidar system and depends on carrier-to-noise ratio and the Doppler peak shape and area. This lead to a discard of an additional 1.8% of the data.

Next, the influence on yaw misalignment estimates of a wake emitted by the Nordtank turbine onto the Vestas V52 is investigated. The derived 10 minute mean misalignment $\widehat{\phi}$ as function of yaw angle of the turbine can be seen in fig. 4.

The vertical black line indicates the position of the Nordtank turbine. It can be seen that outside the wake sector the mis-
alignment ranges around a mean misalignment of $5°$. Inside the wake sector large deviations from the mean misalignment can be observed. In situations where the right half of the rotor (at yaw position $< 195°$) LOS 2 will be affected by a reduced wind speed. Thus, a negative influence on the misalignment estimates by the lidar can be observed. Deviations of up to $30°$ can be

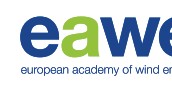
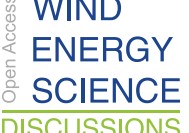


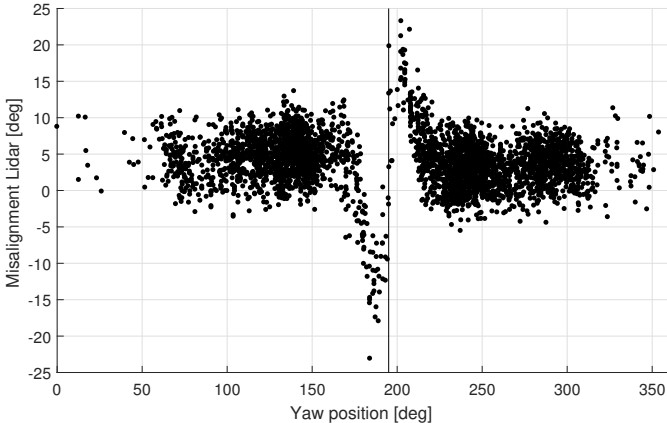

**Figure 4.** Lidar estimated 10 minute mean yaw misalignment $\widehat{\phi}$ as function of turbine yaw position. The vertical black line indicates the position of the upstream Nordtank turbine. A clear influence of the wake onto the misalignment estimates around the wake sector can be seen.

seen. Similar behaviour can be recognized when the wake affects the left part of the rotor (at yaw position $> 195°$). Here, the deviation is smaller, but a clear influence is still visible.

An example of the two spectra are shown in fig. 5. The red spectrum was measured in the free flow and shows a higher wind speed and a slender peak compared to the blue spectrum which is affected by a wake. The reduced wind speed and
5    increased turbulence inside the wake leads to a lower LOS speed and a larger width of the Doppler spectrum compared to the red spectrum.

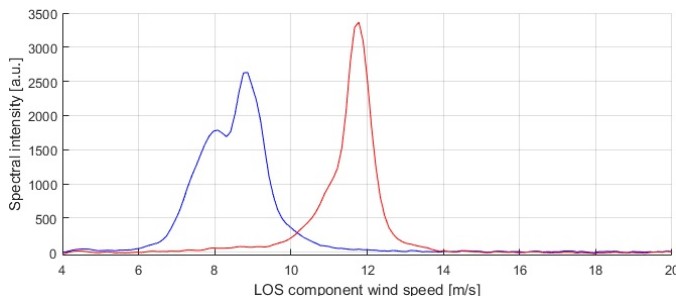

**Figure 5.** Example of two lidar Doppler spectra. The red spectrum is measuring in the free flow, while the blue spectrum is affected by a wake. For the blue spectrum, the reduced wind speed and increased spectral width can clearly be seen. The spectra are averaged over one minute.

A time series result of the detection parameters over three days can be found in fig. 6. The top two panels show $\text{TI}_{\text{LOS}}$ and $\Delta\text{TI}_{\text{LOS}}$, while the bottom panel shows the yaw position of the two turbines. The horizontal black line and the gray shading indicate the wake sector. In general, it can be seen that if both turbines are positioned outside the wake sector, $\text{TI}_{\text{LOS}}$ on both





beams is approximately equal, i.e. $\Delta TI_{LOS} \approx 0$. Thus, no significant difference of the turbulence levels within the lidar probe volumes is detected. As soon as both turbines yaw into a wake situation deviations of $TI_{LOS}$ between beam 1 and 2 can be detected. For example, on 23 December the wind direction changes such that the Vestas V52 is in a half wake situations, where the left half of the rotor is affected based on their yaw positions. In this case a spike in $TI_{LOS1}$ can be identified. This

corresponds to increased small-scale turbulence originating from the wake flow within the probe volume of beam 1 which is not present on beam 2. Hence, a positive increase in $\Delta TI_{LOS}$ can be observed. Similarly, the half-wake situations on the right half of the rotor on 24 December lead to higher $TI_{LOS2}$ than $TI_{LOS1}$.

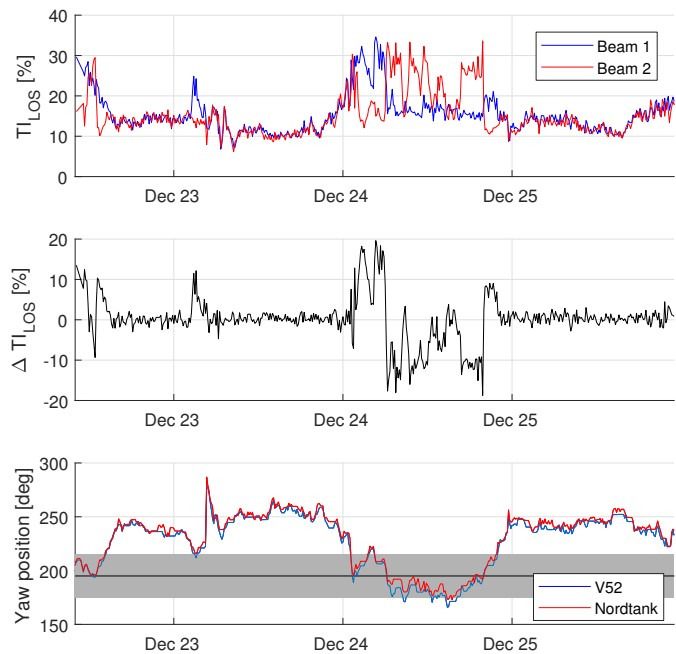

**Figure 6.** Example time series of $TI_{LOS}$ (*top*), $\Delta TI_{LOS}$ (*center*) and the yaw positions of the two turbines (*bottom*). The black horizontal line and the gray shading in the bottom plot indicate the wake position (i.e. $195°$) and a band of $195° \pm 20°$, respectively.

An overview of the measured $TI_{LOS}$ and $\Delta TI_{LOS}$ against the yaw position of the Vestas V52 turbine can be seen in fig. 7. Here, parallel observations to the previous figure can be made. If the turbine is yawing at angles corresponding to half-wake

conditions on the right half of the rotor (i.e $< 195°$) high values for $TI_{LOS2}$ and negative $\Delta TI_{LOS}$ can be observed. This case reverses for half-wake situations on the left half of the rotor where high $TI_{LOS2}$ and positive $\Delta TI_{LOS}$ can be noticed. For the full-wake case increased $TI_{LOS}$ on both beams appears and $\Delta TI_{LOS}$ shows values that are comparable to non-wake, ambient flow conditions. It can also be seen that for non-wake cases $\Delta TI_{LOS}$ is slightly positive. This is due to the mean misalignment which reduces the LOS speed on beam 1, see left illustration in fig. 1. Thus, the ratio $TI_{LOS1}$ is slightly increased compared to

$TI_{LOS2}$.




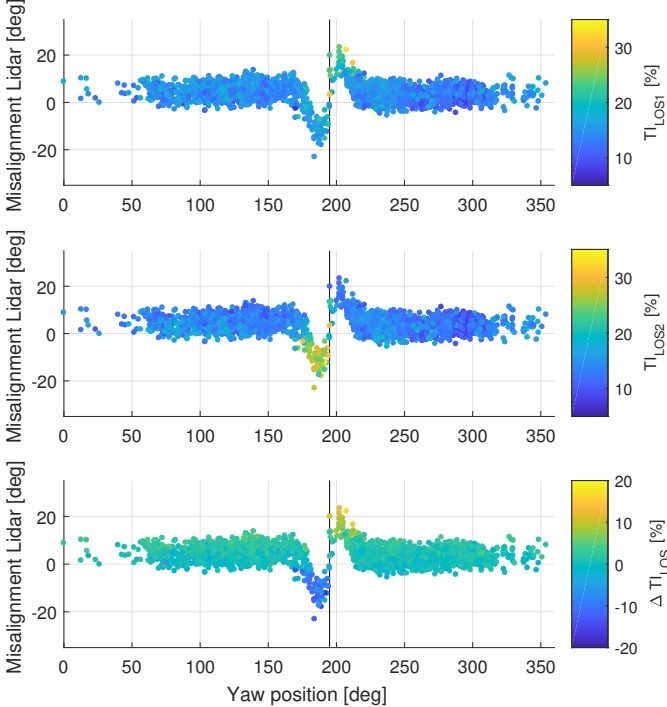

**Figure 7.** Lidar estimated 10 minute mean yaw misalignment $\widehat{\phi}$ as function of turbine yaw position. *Top*: $\text{TI}_{\text{LOS1}}$, *center*: $\text{TI}_{\text{LOS1}}$ and *bottom*:$\Delta\text{TI}_{\text{LOS}}$. The vertical black line indicates the position of the upstream Nordtank turbine (at $195°$).

The next figure presents the results of the detection algorithm, see fig. 8. Again, the misalignment derived from the lidar measurements is plotted against the yaw position of the Vestas turbine. Periods where a wake is detected in the inflow are shown as colored dots and distinguished as left half-wakes (beam 1 is affected by a wake), right half-wakes (beam 2 is affected) and full-wakes where both beams measure a wake influence. It can be seen that all detection results are clustered around the position of the upstream Nordtank turbine where a wake is expected. Also, all negative deviations from the mean misalignment are identified as right half-wakes and all positive deviations as left half-wakes. The recognized full-wakes lie very close to the vertical black line and show no significant discrepancy from the mean misalignment. However, only three full-wake situations were identified. This can be explained by the different rotor sizes and the position of the focus points of the lidar, which are located at towards the edges of the Vestas V52 rotor.

Next, the detected results will be analyzed in detail and the misalignment will be corrected with the help of the undisturbed wind direction measurement from a sonic anemometer mounted on the close-by meteorological mast at the hub height of the Vestas V52 turbine, see fig. 3. First, the correlation between the wind direction estimation from the lidar and the measurement from the sonic will be compared. The absolute wind direction can be obtained by adding the lidar misalignment measurement to the yaw position of the turbine. Both the turbine and the sonic anemometer have been carefully aligned to north. The resulting correlation can be seen in fig. 9.

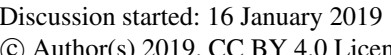



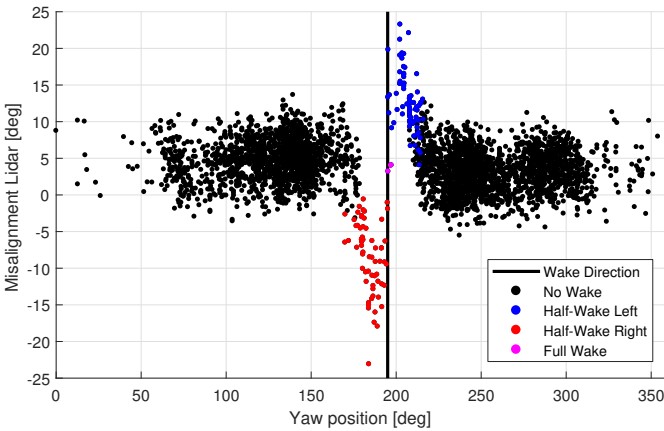

**Figure 8.** Lidar estimated 10 minute mean yaw misalignment $\widehat{\phi}$ as function of turbine yaw position. The vertical black line indicates the position of the upstream Nordtank turbine. Detected waked situations are shown as colored dots.

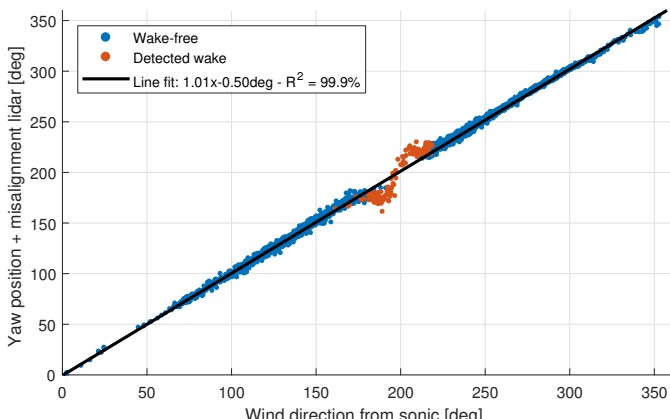

**Figure 9.** 10 minute mean wind direction correlation between the meteorological mast sonic anemometer and the wind direction estimate from the lidar, i.e. lidar misalignment + yaw position of the turbine. Outside the wake sector a high degree of similarity can be observed. The identified wake situations lead to a bias. The line fit is performed on wake-free data only.

It should be noted that outside the wake sector the two signals show a high degree of similarity. The fitted line, which has been fit using the wake-free observations only, shows a unity slope with a very small offset of $0.5°$. The root mean squared error is $2.53°$. However, the measurements which are affected by a wake show the characteristic deviations that could also be seen in fig. 4. This suggest that the flow assumption that are used to derive the lidar misalignment are valid if no wakes are impinging the field of view, but that a correction is necessary for wake situations. If the correlation, which is observed for non-wake observations, also holds for wake cases, the difference between the wind direction from sonic anemometer and lidar can only originate from the influence of the wake.

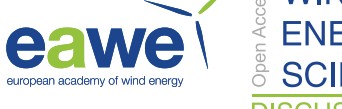

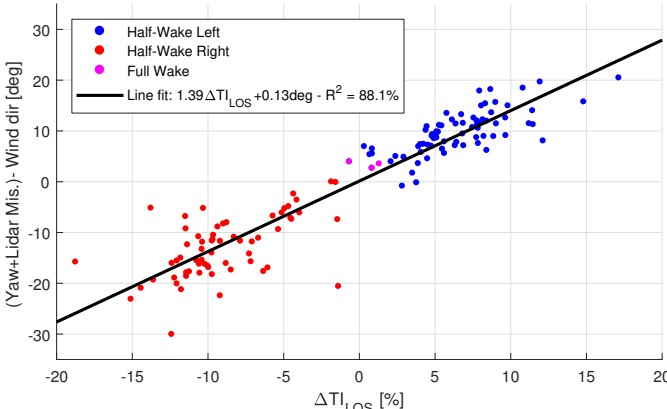

**Figure 10.** Difference between absolute wind direction estimated from the lidar (i.e. lidar misalignment + turbine yaw position) and sonic anemometer versus the detection parameter $\Delta\text{TI}_{\text{LOS}}$. Only the observations where a wake has been identified are shown. The line fit indicates that the data fits well to a linear regression model.

Thus, we suggest to use the wind direction measurements from the sonic anemometer to obtain an empirical relationship between the detection parameter $\Delta\text{TI}_{\text{LOS}}$ and the bias in the wind direction measurements from the lidar and the sonic anemometer. This relationship is shown in fig. 10. It can be observed that detected half wake situations on the right side of the rotor show persistent negative $\Delta\text{TI}_{\text{LOS}}$ and vice versa. The identified full-wakes boh have a wind direction error and $\Delta\text{TI}_{\text{LOS}}$

close to zero.

A linear line has been fitted to the data which shows that the data follows approximately a linear relationship. The derived equation is: $1.39\Delta\text{TI}_{\text{LOS}} + 0.13°$. This equation can now be used to correct the misalignment measurements by the lidar that are affected by a wake. Wake-free observation remain unchanged. The result of the correction can be seen in fig. 11 and by comparing this figure with fig. 8 the performance of the correction method can be evaluated. After the correction it is seen that

the misalignment estimates, which are affected by wakes and previously showed large deviations from the mean misalignment, are now within the range of the wake-free observations. It was possible to adjust both the positive and negative spikes stemming from half-wakes on the left and right side of the rotor. The full-wake situation are only affected slightly because their difference in $\text{TI}_{\text{LOS}}$ is small.

Finally, an overview of the misalignment measurement is compared to the misalignment measured by the sonic anemometer

at the meteorological mast. The misalignment versus wind speed is shown in fig. 12. It can be seen that both the lidar and sonic measurements follow a similar trend. At low wind speeds the misalignment is centered around $5°$ and as the wind speed increases the misalignment reduces. The influence of wakes on the lidar measurements has been identified and corrected as previously described. Slightly lower wind speed measurements from the sonic anemometer can be observed which are caused by the wake of the Vestas V52 turbine on the meteorological mast (at yaw angles of $289°$).




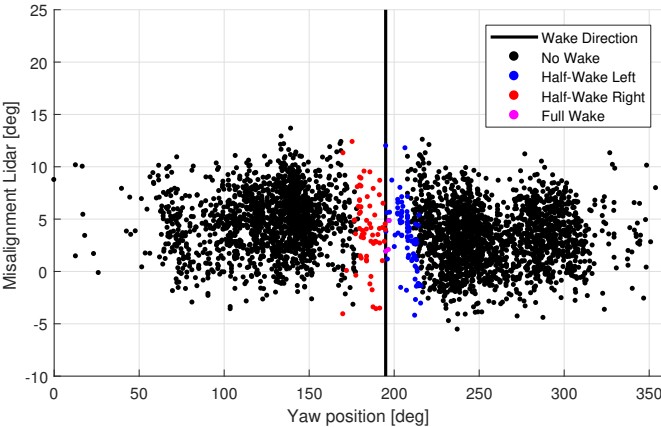

**Figure 11.** Lidar estimated 10 minute mean yaw misalignment $\widehat{\phi}$ as function of turbine yaw position. The vertical black line indicates the position of the upstream Nordtank turbine. Detected wake situations have been corrected according to the relationship found in fig. 9.

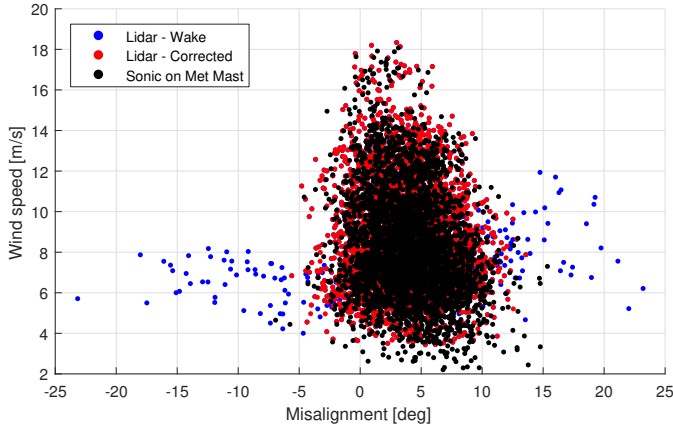

**Figure 12.** Comparison between the 10 minute mean yaw misalignment estimated from the lidar and the sonic on the meteorological mast. The observations affected by wakes are shown in blue and the corrected measurements in red.

## 5 Conclusions

In this study we investigate how nacelle-mounted cw lidar systems can be used to estimate wind turbine misalignment even in inflow with a wake. Lidars offer the possibility to remotely sense the inflow of turbines and avoid the flow disturbance caused by the blades and the nacelle usually encountered by nacelle wind vanes or sonic anemometers. Correct alignment of turbines 5 is important because power production losses can be mitigated.

Nevertheless, due to the limitation of measuring only the LOS component of the approaching wind, a lidar system needs to measure at several positions in front of the wind turbine and assumptions about the flow field need to be made to derive quantities of interest. A common assumption is that of horizontal homogeneity of the inflow which states that at positions of





the same height the wind vector is equal. This assumption is usually satisfied if the terrain is flat and homogeneous and if appropriate averages over time, commonly one to 10 minute averages, are taken. It can, however, be violated if a wake from a neighboring turbine is impinging the inflow and the reduced wind speed inside a wake leads to a bias on the horizontal wind speed and wind direction.

Here, this influence is evaluated experimentally from two measurement campaigns, where a 2-beam cw nacelle lidar was mounted on a Vestas V52 turbine which is exposed to a wake from a slightly smaller neighboring turbine. It was shown that within the wake sector the influence of wakes induces biases on the the misalignment measurement as large as $30°$ from the mean misalignment outside the wake sector. Half-wake situations on the right side of the rotor lead to negative deviations, while half-wakes on the left side result in positive bias. This implies that if lidars were to be used for turbine yaw alignment

the observations that are affected by wake interaction must be identified and corrected.

The wake detection algorithm presented here is based on the spectral broadening effect of the lidar Doppler spectrum because of small-scale turbulence within the probe volume. Since lidar systems perform measurements over a rather large measurement volume, high frequent turbulent fluctuations are attenuated and are not visible in the LOS speed signal, but lead to a widening of the Doppler spectrum. The small-scale turbulence generated inside wake flows will lead to more broadening than in the

ambient flow. Thus, by comparing the spectral width of the Doppler spectrum at different focus locations, wakes that affect the field of view of one or both beams can be detected. The detection parameter used in this study is the LOS-equivalent turbulence intensity $\text{TI}_{\text{LOS}} = \frac{\sigma^2_{\text{LOS}}}{v_{\text{LOS}}}$.

The performance of the algorithm is presented in fig. 8 and shows that all lidar observations that measure the wind direction wrongly were identified. Only very few full-wake situations were observed due to the size difference of the turbines. To correct

the wind direction measurements affected by wakes an empirical relationship between lidar turbulence measurements and the deviation of the lidar wind direction from the true direction measured by a sonic on a nearby mast was established. It was shown that the absolute wind direction measurements from both sensors show a high degree of correlation for non-wake cases. A linear relationship between the difference in $\text{TI}_{\text{LOS}}$ between the two beam and the difference in wind direction between the lidar and the sonic was found. Applying this relationship to the measurements affected by wakes yielded a correction of the

large misalignment deviations experienced during half-wake situations.

Thus, we have shown how the detrimental effect of wake on nacelle lidar measurements can be mitigated. For the correction it was however necessary to utilize the undisturbed wind direction measurements from a mast-mounted sonic anemometer. The turbines that have been used in this experiment are smaller than current utility-scale turbines. Hence, it was possible to use short lidar focus distances which result in a narrow probe volume and the spectral widening is due to turbulence of very small

length scale. In this experiment average spectra of one minute were sufficient to detect the wake-induced turbulence. If the lidar is installed on larger turbine, where larger focus distances are required, the probe volume will increase and shorter averaging times might be necessary to only detect turbulence generated within the wake. Shorter averaging times lead to increased signal noise and can have an effect on the estimation of the Doppler spectrum variance.



*Author contributions.* Dominique P Held performed the research work and prepared the manuscript. Jakob Mann conceived the research plan, supervised the research work and the manuscript preparation.

*Competing interests.* The work of Dominique P Held was partly funded by Windar Photonics A/S in form of an industrial PhD stipend (project number: 5016-00182).

5    *Acknowledgements.* This study was supported by Innovationsfonden Danmark in form of an industrial PhD stipend (project number: 5016-00182). The authors wish to thank Nikolaos Kouris and Antoine Larvol for their contributions to the wake detection algorithm. Further, the authors want to acknowledge the EUDP project *Lidar Detection of Wakes for Wind Turbine Optimization* (project number: 64016-0020).





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
