# Peer review of "Detection of wakes in the inflow of turbines using nacelle lidars"

_Wind Energy Science, 2018_

## Referee Comment (RC1) · Anonymous Referee #1 · 15 Feb 2019

Dear authors,

Thank you very much for submitting your paper to WES journal. I have read it carefully and would like to comment on it.
I recommend to the editor to ask for mayor changes because of the two aspects: 1) the first part of your work have been published before and 2) I question the scientific approach of fitting data to calculate a correction and not generalising and assessing the topic from a scientific point of view.

I had fun reading the first part of your work describing the methodology of detecting wakes with lidar. This is a strong work and worth of publishing. However I realised that this content has been published before, as you indicated on the webpage. In you paper you did not refer to your previous publication, why?

The weak point of your work is the correction. You claim that you need to correct because your assumptions are not valid anymore or phrasing it differently your estimation model is not valid anymore. You phrase your paper as if this is a general problem, however in my point of view this is very specific to the lidar system you use. Let me make two points: 1) a more scientific approach would be to rethink the assumptions of the estimation and change the estimation model/assumptions whenever a wake is detected. 2) a linear data fitting approach is not very novel. You measure a model mismatch fit a correction and your data look nice. I would put a higher standard on a journal publication. Also your simulation work on page 5 is weak. You could introduce a real simulation environment for measuring in a wake in the paper to encourage others to repeat the work. This would give your detection methodology work much more meaning.

I suggest either: to rework the part where you take the information of a wake detection and try to find a better methodology and/or wind model to estimate well even though there is a wake/partial wake;
Or: to focus more on the data analysis and work with the lidar data. Then I suggest to rephrase the title to put the focus on the data evaluation of the field experiment.

**Other points**

| page | line | Comment |
|------|------|---------|
| 1 | 4 | Sufficient large time - statement without a proof / partly disagree |
| 1 | 5 | You don't tell that you would like to measure with Lidar. What about the anemometer or the vane? Are they disturbed, too? |
| 1 | 9 | "one or more beams can be identified" unclear phrasing |
| 1 | 12 | "and thus, correcting the Lidar derived wind direction" see general comment. In my point of view the approach is wrong to correct a estimation which comes from non valid assumptions? |
| 2 | 16 | "as a consequence" - unclear |
| | | Figure 1 variables are not introduced close to the |

| 3 | 1-3 | Terrain effects are present. With them, the same idea can be followed |
|---|---|---|
| 3 | 26 | Please also name the multi distances for the Leosphere device |
| 5 | U_hat | Please introduce before naming it for the first time |
| 7 | - | Chapter 2: What happens to Phi_hat when a wake is detected? How does it effect the algorithm? What happens if a wind direction change and a wake impingement happen simultaneously? |
| 9 | Fig 5 | Does it always look like that? How sensitive is the method? |
| 12 | Fig 9 | Nice evaluation, however it uses the sonic. The transition to a general met mast free methodology is missing. |
| 13 | 1-13 | Fitting of the correction. See comments above. |
| | | Conclusions: What are the learning objectives? Which conclusions can we draw and is there a way to make it independent of the met mast. Is is just for the 2 beam single distance an issue, or for every Lidar system? |

---

## Referee Comment (RC2) · Anonymous Referee #2 · 20 Feb 2019

General Comments:

The paper discusses one of the problems that can arise when measuring wind direction using a nacelle lidar, that the assumption of horizontal homogeneity used to derive the wind direction is not valid when a wake impinges some of the lidar measurement points. However, using a novel approach, the authors discuss how the spectral broadening in the Doppler spectra from the lidar measurements due to small scale wake turbulence can be used to identify measurement periods when wakes are present. Using an empirical correction method based on the measured spectral broadening, the wind direction during waked periods can be corrected.

The paper is well written overall and explains the novel algorithm relatively clearly. However, I believe further discussion on two topics described below should be provided

[Figure]

to strengthen the paper. Additional specific comments are provided as well.

One area I feel could use more discussion in particular is missed detections and false alarms. What is the probability of a false detection of wake impingement from the experiment, and the same for missed detections when wake impingement actually occurred? How were these probabilities accounted for when deciding which thresholds to use for the detection algorithm?

The other area that I believe should be discussed more is the applicability of the algorithm to different wind turbines, sites, and wind conditions. This research demonstrates that the LOS Doppler spectrum can be used to detect waked conditions well for the site and conditions analyzed. Although briefly discussed in the conclusions, it is unclear what steps would need to be taken to implement the method at a different site with a different rotor size, turbine spacing, or atmospheric conditions. For example, simulations of the algorithm for different conditions using CFD would be a useful approach. Further analysis of the wind conditions during the experiment, such as turbulence intensity and atmospheric stability, would help show how applicable the algorithm is to a variety of wind conditions.

Specific Comments:

Section 1: There could be value in identifying when a turbine is waked for purposes such as wind farm control, but this is not discussed much in the paper. Do you have any ideas about the potential usefulness of the algorithm in wind farm control strategies?

Pg. 5, ln. 18: "The effect is strongest for negative turbine misalignments" From Fig. 2 it appears that the impact of increasing wake deficits on the measured direction bias is roughly equal for all misalignments. Can you explain this statement further?

Pg. 5, ln. 18: "The kinks that appear for the negative turbine misalignments" In Fig. 2, it appears that the kinks are for some "positive" misalignments.

Pg. 6, ln. 20: "high frequency components of the streamwise component increased

fourfold in the wake" How far behind the turbine was this increase found?

Eq. 12: By using the LOS Doppler spectrum TI to detect wake impingement, what would happen if a naturally occurring gust was present on one side of the rotor but not the other? Even if the turbulence is the same at the two beams, the lower velocity at one beam would cause an increase in TI, which could trigger a wake detection.

Pg. 7, ln. 2: "calculated from one minute average spectra." Did you look at the sensitivity to different averaging times, and how did you settle on one minute?

Pg. 7, ln. 11: "At the initialization the algorithm requires some observations to establish correct values of the running averages." Explain in more detail how the initialization of the algorithm is performed. Does the algorithm require that the wind conditions during operation be similar to the conditions during initialization? And how frequently does the algorithm need to be calibrated? Especially for detecting full wake conditions when the absolute TI is used to detect wakes, how do you account for the possibility of the freestream TI increasing after the algorithm is initialized, in which case higher freestream TI could be detected as a full wake?

Fig. 5: How comparable are the wind conditions for these two spectra? For example, was the freestream TI the same for both periods, so that the difference should be due to the impact of the wake? Some further discussion would be appreciated.

Pg. 13, lns. 1-5: Does the empirical relationship used to correct wind direction measurements when wakes are detected need to be determined for every site where the algorithm is used? Or is the relationship found valid in general? Additionally, after correcting the wind directions, how does the RMS error between the corrected lidar wind direction and the sonic anemometer compare to the error during freestream conditions? Although the corrected directions look reasonable, some quantification of the error would strengthen the results.

Technical Corrections:

Pg. 6, ln. 28: "From both properties mentioned above" Which properties are being referenced here?

Pg. 7, ln. 17: "Further, data from a meteorological mast at a distance of 120 m..." Incomplete sentence

Pg. 8, ln. 16: "In situations where the right half of the rotor ..." Check grammar in this sentence.

Pg. 10, ln. 11: "TI_LOS2" -> "TI_LOS1"?

―――――――――――――――――――

---

## Author Comment (AC1) · 28 Mar 2019

The authors response to RC1 and RC2 can be found in the attached documents. The changes to the manuscript can be found in diff.pdf.

Please also note the supplement to this comment:
https://www.wind-energ-sci-discuss.net/wes-2018-78/wes-2018-78-AC1-supplement.zip

---

## Author Response (AR1)

**Author Response to Review Comment #1**

Dear Reviewer,

Thank you for reviewing the manuscript. Your comments were very helpful and improved the quality of the manuscript. The author responses can be found below each reviewer comment.

| | |
|---|---|
| RC 1.1 | This is a strong work and worth of publishing. However I realised that this content has been published before, as you indicated on the webpage. In you paper you did not refer to your previous publication, why? |

AC   Results of the wake detection algorithm have been published before. However, these results came from a different experiment at a different wind farm. The correction method has not been published before. A paragraph referencing to the already published results have been added to the introduction section.

| | |
|---|---|
| RC 1.2 | The weak point of your work is the correction. You claim that you need to correct because your assumptions are not valid anymore or phrasing it differently your estimation model is not valid anymore. You phrase your paper as if this is a general problem, however in my point of view this is very specific to the lidar system you use. Let me make two points: 1) a more scientific approach would be to rethink the assumptions of the estimation and change the estimation model/assumptions whenever a wake is detected. 2) a linear data fitting approach is not very novel. You measure a model mismatch fit a correction and your data look nice. I would put a higher standard on a journal publication. Also your simulation work on page 5 is weak. You could introduce a real simulation environment for measuring in a wake in the paper to encourage others to repeat the work. This would give your detection methodology work much more meaning. |

AC   The problem of a violation of the assumption of horizontal homogeneity is relevant for many lidar applications (not only nacelle lidars). Here we propose a method that can be applied to all nacelle-mounted continuous-wave lidars.
We present an algorithm that is able to identify situations where the assumptions of yaw misalignment estimation by nacelle lidars are violated. We believe that it is a plausible approach to apply a correction to the estimation, when the assumptions are violated.
We agree that the linear data fitting approach is not very novel. However, it is applied in many areas of wind energy. We show that the linear model agrees well with data gathered during *this experiment* and thus proceed to use it for the correction of the misalignment measurements. For an application of the method to other experiments more data is needed, where the method can be tested. Our goal in this work is to show the results of one setup, where this method works.
The simulation work on page 5 should give the reader an illustration of how severe the effect of wakes on the estimation of horizontal wind speed and wind direction can be.

RC 1.3    p. 1 l. 4: Sufficient large time - statement without a proof / partly disagree

AC    It was shown in many previous studies that the assumption of horizontally homogeneous flow is valid in flat homogeneous terrain and when performing sufficiently large time averages. Also in this work agreement between reference sensors and the lidar is demonstrated.

RC 1.4    p. 1 l. 5: You don't tell that you would like to measure with Lidar. What about the anemometer or the vane? Are they disturbed, too?

AC    A nacelle sonic anemometer or a wind vane are disturbed by the flow around the rotor and the nacelle. This disturbance is mentioned more detailed in the introduction section. For the lidar system, the disturbance stems from the violation of horizontally homogeneous flow, which can be caused by wakes from neighboring turbines and is thus of different nature.

RC 1.5    p. 1 l. 9: "one or more beams can be identified" unclear phrasing

AC    This has been rephrased. Now "measurement locations" is used instead of "beams".

RC 1.6    p. 1 l. 12: "and thus, correcting the Lidar derived wind direction" see general comment. In my point of view the approach is wrong to correct a estimation which comes from non valid assumptions?

AC    Again, we believe it is a reasonable approach. See also the response to the previous comment RC 1.2.

RC 1.7    p. 2 l. 16: "as a consequence" – unclear

AC    This has been rephrased.

RC 1.8    p. 3 l. 1-3: Figure 1 variables are not introduced close to the Terrain effects are present. With them, the same idea can be followed

AC    A reference to the page where the variables are defined has been added.

RC 1.9    p. 3 l. 26: Please also name the multi distances for the Leosphere device

AC    The multiple focus distances have been added.

RC 1.10   p. 5 U_hat, Please introduce before naming it for the first time

AC    The hat notation is defined on p. 5 l. 3 and the variables are defined on p. 5 l. 10.

RC 1.11   p. 7: Chapter 2: What happens to Phi_hat when a wake is detected? How does it effect the algorithm? What happens if a wind direction change and a wake impingement happen simultaneously?

AC  Phi_hat will show large deviations if a wake is affecting on of the measurement locations. This is illustrated for measured data in fig. 4. If a wake is detected, the estimation of Phi_hat will not change unless the proposed correction method is used.
A running average of the wake-free detection parameters is calculated and compared to the instantaneous values. Thresholds then decided whether a wake is detected or not. This has been added to the end of section 2.
A wind direction change can coincide with a wake impingement and in such a case the algorithm cannot distinguish these events. This comment has been added at the end of section 2.

RC 1.12   p. 9 fig. 5: Does it always look like that? How sensitive is the method?

AC  This is an example of two spectra measured during a wake situation. This should illustrate the broadening of the Doppler spectra due to increased small-scale turbulence. In wake situations the spectra will typically look like this. The method is sensitive to the averaging time. We have tried several averaging times and found that 1 minute averages give the best results.

RC 1.13   p. 12 fig. 9: Nice evaluation, however it uses the sonic. The transition to a general met mast free methodology is missing.

AC  For a mast-free correction method more data is need to understand how the linear relationship changes with site specific parameter, e.g. distance between turbines, rotor diameter, ambient turbulence levels. We have added a paragraph at the end of the conclusion section, where this issue is discussed.

RC 1.14   Conclusions: What are the learning objectives? Which conclusions can we draw and is there a way to make it independent of the met mast. Is is just for the 2 beam single distance an issue, or for every Lidar system?

AC  Similarly to the previous comment, more details have been added at the end of the conclusion section. The issues arises for all nacelle-based continuous-wave lidars that focus at different positions at a fixed focus distance.

**Author Response to Review Comment #2**

Dear Reviewer,

Thank you for reviewing the manuscript. Your comments were very helpful and improved the quality of the manuscript. The author responses can be found below each reviewer comment.

RC 2.1    One area I feel could use more discussion in particular is missed detections and false alarms. What is the probability of a false detection of wake impingement from the experiment, and the same for missed detections when wake impingement actually occurred? How were these probabilities accounted for when deciding which thresholds to use for the detection algorithm?

AC    Currently we cannot make any statements about the probability of a false detection. The purpose of the manuscript is to introduce the method to the community and show that it gives satisfactory results for the experiment considered here.

RC 2.2    The other area that I believe should be discussed more is the applicability of the algorithm to different wind turbines, sites, and wind conditions. This research demonstrates that the LOS Doppler spectrum can be used to detect waked conditions well for the site and conditions analyzed. Although briefly discussed in the conclusions, it is unclear what steps would need to be taken to implement the method at a different site with a different rotor size, turbine spacing, or atmospheric conditions. For example, simulations of the algorithm for different conditions using CFD would be a useful approach. Further analysis of the wind conditions during the experiment, such as turbulence intensity and atmospheric stability, would help show how applicable the algorithm is to a variety of wind conditions.

AC    A paragraph at the end of the conclusion section has been added that discusses the topic of different site parameters. We believe that more data is needed to make statements about the generality of the linear relationship found in this experiment.

RC 2.3    Section 1: There could be value in identifying when a turbine is waked for purposes such as wind farm control, but this is not discussed much in the paper. Do you have any ideas about the potential usefulness of the algorithm in wind farm control strategies?

AC    Yes, we have added a paragraph at the end of the introduction section that discusses the idea of using wake detection results for wind farm control.

RC 2.4    Pg. 5, ln. 18: "The effect is strongest for negative turbine misalignments" From Fig. 2 it appears that the impact of increasing wake deficits on the measured direction bias is roughly equal for all misalignments. Can you explain this statement further?

AC    This statement has been removed.

> RC 2.5   Pg. 5, ln. 18: "The kinks that appear for the negative turbine misalignments" In Fig. 2, it appears that the kinks are for some "positive" misalignments.

AC    Yes, this has been corrected.

> RC 2.6   Pg. 6, ln. 20: "high frequency components of the streamwise component increased fourfold in the wake" How far behind the turbine was this increase found?

AC    This was at three rotor diameters behind the turbine and has been added to the text.

> RC 2.7   Eq. 12: By using the LOS Doppler spectrum TI to detect wake impingement, what would happen if a naturally occurring gust was present on one side of the rotor but not the other? Even if the turbulence is the same at the two beams, the lower velocity at one beam would cause an increase in TI, which could trigger a wake detection.

AC    This observation is correct. However, we believe that the occurrence of a gust that will only affect one measurement location on averaging time of one minute is quite unlikely. We have not seen evidence of such an event in the data analyzed here.

> RC 2.8   Pg. 7, ln. 2: "calculated from one minute average spectra." Did you look at the sensitivity to different averaging times, and how did you settle on one minute?

AC    Yes, we have also tried shorted averaging times, but we concluded that one minute averages gave the best performance. This fact has been added to the text.

> RC 2.9   Pg. 7, ln. 11: "At the initialization the algorithm requires some observations to establish correct values of the running averages." Explain in more detail how the initialization of the algorithm is performed. Does the algorithm require that the wind conditions during operation be similar to the conditions during initialization? And how frequently does the algorithm need to be calibrated? Especially for detecting full wake conditions when the absolute TI is used to detect wakes, how do you account for the possibility of the freestream TI increasing after the algorithm is initialized, in which case higher freestream TI could be detected as a full wake?

AC    More details to the initialization have been added.
Yes, the conditions during operation need to be similar to conditions during operations. This fact has been added to the text.
From our experience no calibration is need with the data we have processed.
The algorithm assumes that all changes in small-scale turbulence stem from upstream wakes.

RC 2.10 Fig. 5: How comparable are the wind conditions for these two spectra? For example, was the freestream TI the same for both periods, so that the difference should be due to the impact of the wake? Some further discussion would be appreciated.

AC The two spectra were measured during the same one minute interval and thus the ambient turbulence conditions can be assumed equal. This has been added to the text.

RC 2.11 Pg. 13, lns. 1-5: Does the empirical relationship used to correct wind direction measurements when wakes are detected need to be determined for every site where the algorithm is used? Or is the relationship found valid in general? Additionally, after correcting the wind directions, how does the RMS error between the corrected lidar wind direction and the sonic anemometer compare to the error during freestream conditions? Although the corrected directions look reasonable, some quantification of the error would strengthen the results.

AC To validate the generality of the linear relationship more empirical data is needed. This is mentioned at the end of the conclusion section and was also mentioned by RC1 in comment RC 1.13 & 1.14. We currently have some indication that the relationship at different site is also well described by a linear fit. However, the offsets is close to zero (as found in this study), but the slope seems to vary.

RC 2.12 Pg. 6, ln. 28: "From both properties mentioned above" Which properties are being referenced here?

AC This refers to the numbered list at the beginning of the subsection. A more clear formulation has been used.

RC 2.13 Pg. 7, ln. 17: "Further, data from a meteorological mast at a distance of 120 m..." Incomplete sentence

AC This sentence has been completed.

RC 2.14 Pg. 8, ln. 16: "In situations where the right half of the rotor ..." Check grammar in this sentence.

AC This sentence has been completed.

RC 2.15 Pg. 10, ln. 11: "TI_LOS2" -> "TI_LOS1"?

AC Yes, this has been corrected.

[revised manuscript text omitted]

---

## Author Response (AR2)

Dear reviewer,

We thank the reviewer for the additional comments which we believe have improved the manuscript. Please find the author responses in the text below.

1) "RC 2.1: One area I feel could use more discussion in particular is missed detections and false alarms. What is the probability of a false detection of wake impingement from the experiment, and the same for missed detections when wake impingement actually occurred? How were these probabilities accounted for when deciding which thresholds to use for the detection algorithm?"

AC: "Currently we cannot make any statements about the probability of a false detection. The purpose of the manuscript is to introduce the method to the community and show that it gives satisfactory results for the experiment considered here."

RC 2.1 : Although you might not be able to determine probabilities, please discuss whether there are any occurrences of missed detections and false alarms in the data. For example, in the wake sector, are there any periods not flagged as waked by the algorithm? And if so, how many? Same for the detection of waked periods outside of the wake sector.

AC: Yes, we have included a discussion on the occurrences of missed and false detection. In general, no false detections occur outside the wake sector and no detections are missed inside the wake sector.

2) RC 2.2 "The other area that I believe should be discussed more is the applicability of the algorithm to different wind turbines, sites, and wind conditions. This research demonstrates that the LOS Doppler spectrum can be used to detect waked conditions well for the site and conditions analyzed. Although briefly discussed in the conclusions, it is unclear what steps would need to be taken to implement the method at a different site with a different rotor size, turbine spacing, or atmospheric conditions. For example, simulations of the algorithm for different conditions using CFD would be a useful approach. Further analysis of the wind conditions during the experiment, such as turbulence intensity and atmospheric stability, would help show how applicable the algorithm is to a variety of wind conditions."

AC "A paragraph at the end of the conclusion section has been added that discusses the topic of different site parameters. We believe that more data is needed to make statements about the generality of the linear relationship found in this experiment."

RC 2.2.: The new paragraph explains that further experiments are needed to determine how general the wind direction correction algorithm is, but the generality of the wake detection algorithm should be discussed as well. Furthermore, some site condition information such as the distribution of wind speed and direction is provided in Fig. 3, but can you add information at least about the distribution of TI and

ideally stability during the measurement period? You show that the algorithm works well for this period, but knowing what conditions occurred during this period will help readers understand the general applicability of the algorithm.

AC: We have added a plot of the TI and stability of the atmosphere during the experiment. We also added a paragraph that discussed the result in view of the TI and stability.

3) RC 2.9 "Pg. 7, ln. 11: "At the initialization the algorithm requires some observations to establish correct values of the running averages." Explain in more detail how the initialization of the algorithm is performed. Does the algorithm require that the wind conditions during operation be similar to the conditions during initialization? And how frequently does the algorithm need to be calibrated? Especially for detecting full wake conditions when the absolute TI is used to detect wakes, how do you account for the possibility of the freestream TI increasing after the algorithm is initialized, in which case higher freestream TI could be detected as a full wake?"

AC "More details to the initialization have been added. Yes, the conditions during operation need to be similar to conditions during operations. This fact has been added to the text. From our experience no calibration is need with the data we have processed. The algorithm assumes that all changes in small-scale turbulence stem from upstream wakes."

RC 2.9: In the new text you mention that running averages of $\hat{\phi}$, TI_LOS2, and TI_LOS1 are needed to initialize the algorithm. But the description of the algorithm for wake detection only mentions using DeltaTI_LOS and TI_LOS1 and TI_LOS2. Do you also use $\hat{\phi}$ as part of the algorithm? If so, provide more details. It would help greatly with understanding the algorithm if you explain how the various detection thresholds for partial wake and full wake situations are chosen based on the running averages.

AC: Currently, we mention the following on the detection algorithm:

"The wake detection algorithm is designed to treat the one minute spectra consecutively and will compare the values of the detection parameters TILOS(rLOS,1), TILOS(rLOS,2), ΔTILOS and $\hat{\phi}$ to their values from ambient, wake-free conditions. A running average of these values is calculated from previous measurements. The running averages aim at representing the values of the detection parameters under ambient, wake-free conditions and situations, where a wake has been identified, will not contribute to the running averages."

and on its initialization:

"At the initialization the algorithm requires some wake-free observations to establish correct values of the running averages. The observations are used to calculate running averages of $\hat{\phi}$,TILOS2 and TILOS1."

We have added that the threshold values have been found manually.

4) RC 2.11 "Pg. 13, lns. 1-5: Does the empirical relationship used to correct wind direction measurements when wakes are detected need to be determined for every site where the algorithm is used? Or is the relationship found valid in general? Additionally, after correcting the wind directions, how does the RMS error between the corrected lidar wind direction and the sonic anemometer compare to the error during freestream conditions? Although the corrected directions look reasonable, some quantification of the error would strengthen the results."

AC "To validate the generality of the linear relationship more empirical data is needed. This is mentioned at the end of the conclusion section and was also mentioned by RC1 in comment RC 1.13 & 1.14. We currently have some indication that the relationship at different site is also well described by a linear fit. However, the offsets is close to zero (as found in this study), but the slope seems to vary."

RC 2.11: How does the RMS error between the corrected lidar wind direction estimate and the sonic wind direction compare to the error during freestream conditions? Has the correction reduced the error to close to the freestream error? Some quantification of the results would be appreciated.

AC: We have added the RMSE to the result section. After the correction the RMSE of the wake-affected data is similar to the wake-free data.

[revised manuscript text omitted]